



**Benthic C fixation and cycling in diffuse hydrothermal and**
**background sediments in the Bransfield Strait, Antarctica**
Clare Woulds*[1], James B. Bell[1, 2], Adrian G. Glover[3], Steven Bouillon[4], Louise S. Brown[1, 2]
[1]water@leeds, School of Geography, University of Leeds, Leeds, LS2 9JT, UK
[2]Cefas, Pakefield Road, Lowestoft, Suffolk, NR33 0HT, UK
[3]Life Sciences Dept., Natural History Museum, Cromwell Rd, London SW7 5BD, UK
[4]Department of Earth and Environmental Sciences, KU Leuven, Leuven, Belgium
*Correspondence to: c.woulds@leeds.ac.uk
**Abstract**
Sedimented hydrothermal vents are likely to be widespread compared to hard substrate hot vents. They host
chemosynthetic microbial communities which fix inorganic C at the seafloor, as well as a wide range of
macroinfauna, including vent-obligate and background non-vent taxa. There are no previous direct observations
of Carbon cycling at a sedimented hydrothermal vent.
We conducted [13]C isotope tracing experiments at 3 sedimented sites in the Bransfield Strait, Antarctica, which
showed different degrees of hydrothermalism. Two experimental treatments were applied, with [13]C added as
either algal detritus (photosynthetic C), or as bicarbonate (substrate for benthic C fixation).
Algal [13]C was taken up by both bacteria and metazoan macrofaunal, but its dominant fate was respiration, as
observed at deeper and more food limited sites elsewhere. Rates of [13]C uptake and respiration suggested that the
diffuse hydrothermal site was not the hotspot of benthic C-cycling that we hypothesised it would be.
Fixation of inorganic C into bacterial biomass was observed at all, and was measurable at 2 out of 3 sites. At all
sites, newly fixed C was transferred to metazoan macrofauna. Fixation rates were relatively low compared to
similar experiments elsewhere, thus C fixed at the seafloor was a minor C source for the benthic ecosystem.
However, as the greatest amount of benthic C fixation occurred at the off vent (non-hydrothermal) site





(0.077±0.034 mg C m$^{-2}$ fixed during 60 h), we suggest that benthic fixation of inorganic C is more widespread
than previously thought, and warrants further study.



## 1. Introduction

Sedimented hydrothermal vent (SHV) sites are those where hydrothermal fluid diffuses through soft sediment
cover on its way to mixing with oceanic bottom water. This creates hot (up to ~100°C) sediments with
porewaters rich in dissolved sulphide and methane, which supports microbes that conduct chemosynthetic C
fixation through a range of pathways (Bernardino et al., 2012). These hydrothermally influenced sediments are
likely to be more spatially extensive than hard substrate vents, although their diffusive nature makes their extent
hard to quantify. Sedimented hydrothermal vents have been shown to influence biological community
composition and nutrition at adjacent sites which were otherwise characterised as 'inactive' or 'off-vent' (Levin
et al., 2009; Bell et al., 2016a; Bell et al. 2016b; Bell et al., 2017a). However, the ecology of sedimented
hydrothermal sites has received relatively little study. There is only one modelling study that has focused on the
interaction between benthic ecosystems and C-cycling at SHVs (Bell et al., 2017b), and there are no direct
observations of SHV C-cycling by components of the benthic ecosystem.
So far, a limited number of studies have used natural stable isotopic analysis to determine carbon sources and
their fixation pathways utilised by infauna at SHVs (Levin et al., 2009; Soto, 2009; Sweetman et al., 2013; Bell
et al. 2016b; Portail et al. 2016). Evidence has shown that C fixed during anaerobic oxidation of methane, oxic
methanotrophy, sulphide oxidation, as well photosynthetic organic matter (OM) sinking from the surface, are all
utilised by macrofauna to varying extents at SHVs (Levin et al., 2009; Bernardino et al., 2012). It is very
challenging however to quantify the relative contributions of different C sources to macrofaunal diets, both
because the natural isotopic ranges of some C sources tend to overlap, and because often the isotopic
compositions of those end members could not be measured (Levin et al., 2009; Bell et al., 2016b). Unknown
variability in trophic discrimination factors also currently preclude quantitative estimates of the relative
contribution of different C sources.
Stable isotope tracing experiments offer a way to overcome some of these issues. The experimental addition of
labelled C sources, either photosynthetic OM or dissolved inorganic C (provided as bicarbonate) to SHV
sediment allows production of chemosynthetic OM, and the transfer of different OM types into the
macrobenthos and other C pools to be directly observed. Such experiments (using only photosynthetic OM)
have been conducted at a wide range of (ostensibly) non-chemosynthetic benthic sites, and have shown a wide
variation in the relative importance of different biological C processing pathways (Woulds et al., 2009; 2016).
At food limited sites in the deep-sea, respiration tends to be the dominant fate of added OM (van Oevelen et al.
2011; 2012). However, shallower, more food rich settings such as coastal fjords and estuaries, with greater





sedimentary organic C concentrations and higher macrofaunal biomass, show a pattern of biological C
processing in which uptake by fauna is a more important process, and at unusual and particularly food rich sites,
such as the lower margin of the Arabian Sea oxygen minimum zone (~1000 m depth), macrofaunal C uptake can
even be the dominant process (Woulds et al., 2009; 2016).
The occurrence of chemosynthesis in a benthic habitat represents an additional source of fresh, labile OM in an
environment that would otherwise be more severely food limited. For this reason, it has been suggested that
hydrothermally influenced sites can be biomass hotspots, where biogeochemical cycling is rapid (Bernardino et
al., 2012). However, due to the environmental toxicity created by hydrothermal fluid, and the fact that the
majority of taxa inhabiting SHVs are background rather than vent-endemic, the difference in faunal biomass
between SHVs and adjacent non-vent sites is highly variable (Levin et al., 2009; Bernardino et al., 2012; Bell et
al., 2016). It therefore seems possible that biological C processing at SHVs will show a distinct complement of
biological C processing patterns unlike those observed elsewhere in the deep sea. The food rich, high biomass
characteristics of some SHVs may lead to biological C processing that is and more similar to shallower, food
rich environments. On the contrary however, spatially variable biomass patterns, as well as the metabolic costs
associated with potentially high temperatures and porewater toxicity would tend to counteract the effect of
enhanced food availability. Therefore overall, as direct measurements of biological C processing rates and
pathways have not previously been made at SHVs or in the Southern Ocean, there remains a gap in our
understanding of sedimentary C and N-cycling.
**1.1 Hypotheses**
In this study we conducted stable isotope tracing experiments at three sites of variable hydrothermal activity in
the Bransfield Strait, Antarctica. To the best of our knowledge this is the first isotope tracing experiment in this
type of system. The following hypotheses were addressed:
• Hydrothermally influenced sites exhibiting chemosynthesis will show elevated rates of biological C
processing.
• At hydrothermally influenced sites inorganic substrate will be fixed by chemoautotrophs and
transferred to the macrofauna.
• Preference for feeding on photosynthetic versus chemosynthetic OM will be taxon dependent.



## 2. Methods

### 2.1 Study sites

In this study we focus on a SHV in the Bransfield Strait, close to the tip of the Antarctic peninsula. The discovery of hydrothermal venting in the Bransfield Strait was reported by Klinkhammer et al. (2001), who detected hydrothermal plumes in the water column, and recovered hot 'soupy' sediment from Hook Ridge. In addition, a new species of *Sclerolinum* (Sahling et al., 2005) there has been described, and porewater geochemistry and hydrothermal flux rates have been published (Sahling et al., 2005; Aquilina et al., 2013).

Experiments were conducted at three sites in the Bransfield Strait, Antarctica (Fig. 1). Two of the sites lay on raised edifices, known as Hook Ridge and Middle Sister, along the axis of the basin, and were selected as being likely to exhibit diffuse hydrothermal venting, and the former was the location where diffuse venting had been identified. A third site, at a similar depth but along the north side of the basin, was chosen as an off-vent control (hereafter known as 'Off-Vent').

Sediment organic carbon (Corg) concentrations were lower at Hook Ridge (0.97 wt% Corg) than at the Off-Vent and Middle Sister sites, which showed similar values (1.35 and 1.4 wt% Corg respectively, Table 1). The sites differed in biomass of different groups, with Hook Ridge and Middle Sister showing higher bacterial biomass and lower macrofaunal biomass than the Off-Vent site (Table 1). Hook Ridge was the only site classified as hydrothermally active by Aquilina et al. (2013), with porewaters enriched in sulphide, methane and dissolved metals and depleted in chloride. Macrofauna tended to be representative of the background taxa of the region. Each site also supported one species of siboglinid polychaete. In the case of Hook Ridge this was *S. contortum*, and at Middle Sister and the Off-Vent site it was *Siboglinum sp.*, and they were always a minority constituent of the community (Bell et al., 2016 a).

### 2.2 Isotope tracing experiments

Sediment cores (10 cm i.d.) were recovered using a multiple corer, and kept in the dark at seafloor temperatures (Table 1) using cooled incubators. Experiments were initiated by addition of isotopically enriched substrates. Cores were then sealed and incubated for ~60 h, during which core-top water was continuously stirred.

Duplicate cores were subjected to each of two treatments. In the 'algae' treatment, marine algal detritus (*Chlorella*, Cambridge Isotope Laboratories) enriched in $^{13}C$ and $^{15}N$ (both ~100 at %) was allowed to settle on the sediment surface, giving a final dose of $436\pm30$ mg C m$^{-2}$. In the 'Bicarbonate' treatment a solution of 100



% $^{13}$C labelled sodium bicarbonate and 100 % $^{15}$N labelled ammonium chloride was injected in the surface 5 cm
of sediment porewater, to give a dose of 306 mg C m$^{-2}$ and 2.52 mg N m$^{-2}$.
At intervals during the incubation, core top water samples were withdrawn from Algae treatment cores, and
stored in crimp-cap vials poisoned with HgCl$_2$ for dissolved inorganic carbon (DIC) analysis. At the end of the
experiment cores were extruded and sectioned at intervals of 0-1, 1-2, 2-3, 3-5 and 5-10 cm. Half of each section
was frozen at -20°C, and the other half was preserved in buffered 10% formalin.
**2.3 Sample processing and analysis**
Overlying water samples were analysed for concentration and isotopic composition of DIC in triplicate on a
Thermalox TOC analyser coupled to a Thermo Delta V Advantage IRMS via a Conflo IV interface, using a
Thermo TriPlus autosampler. The reaction column was filled with H$_3$PO$_4$-coated beads.
Frozen sediment samples were freeze dried and analysed for phospholipid fatty acids (PLFAs) following Main
et al. (2015). Briefly, samples were extracted in a modified Bligh and Dyer extraction solution of
chloroform:methanol:citrate buffer, 1:2:0.8. The polar fraction was obtained by loading samples onto ISOLUTE
SPE columns, washing with chloroform and acetone, and eluting with methanol. After addition of nonadecanoic
acid (C19:0) as an internal standard, extracts were derivatised in the presence of KOH in methanol.
Derivatisation was quenched with water and acetic acid, and the organic fraction was extracted by washing with
4:1 isohexane:chloroform. Samples were dried and then taken up in isohexane for analysis on a Trace Ultra GC,
connected via a GC Combustion III to a Delta V Advantage IRMS (Thermo Finnigan, Bremen). The isotopic
signature of each PLFA was measured against a CO$_2$ reference gas which is traceable to IAEA reference
material NBS 19 TS-Limestone, with a precision of ± 0.31 ‰, and corrected for the C atom added during
derivatization.
Sediment preserved in formalin was sieved over a 300μm mesh. Macrofauna were extracted under a binocular
microscope, identified to broad taxonomic level, air dried in pre-weighed tin capsules, and weighed. In some
cases multiple individuals were pooled to create samples large enough for analysis. Fauna were de-carbonated
by dropwise addition of 0.1M HCl, followed by air drying at 50°C. Calcareous foraminifera and bivalves which
were too small for manual removal of shells were de-carbonated with 6N HCl. Fauna were analysed for their C
contents and isotopic signature using a Flash EA 1112 Series Elemental Analyser connected via a Conflo III to a
Delta$^{Plus}$ XP isotope ratio mass spectrometer (all Thermo Finnigan, Bremen). Carbon contents was quantified
using the area under the mass spectrometer response curve, against National Institute of Standards and



Technology reference material 1547 peach leaves (repeat analysis gave precision ± 0.35 %). Isotopic data were
traceable to IAEA reference materials USGS40 and USGS41 (both L-glutamic acid), with a precision ± 0.13 ‰.

**2.4 Data treatment**

Respiration of added algal C was calculated for cores subjected to the algae treatment. The amount of excess
DI$^{13}$C in each sample was calculated by first subtracting the natural abundance of $^{13}$C in DIC. This was scaled
up to give the total amount of DIC from the added algae at each sample timepoint, and corrected for water
removed and added during sampling. Respiration rate was calculated for each core by placing a line of best fit
through the amount of added $^{13}$C over time, and normalised to surface area.
Bacterial incorporation of $^{13}$C was calculated by first subtracting the natural abundance of $^{13}$C from the isotopic
signature of each PLFA (data published in Bell et al., 2017), to give the amount of added C in each compound.
Bacterial incorporation was then calculated using the 4 bacteria-specific PLFAs isoC14:0, isoC15:0,
antisoC15:0, and isoC16:0, following Boschker and Middelburg (2002). Uptake of $^{13}$C into these bacteria-
specific PLFAs was summed, and scaled up on the basis that they together account for 14% of total bacterial
PLFA, and that PLFAs account for 5.6% of total bacterial biomass. For samples in the bicarbonate treatment
further scaling up was applied, to account for the fact that the addition of $^{13}$C bicarbonate was calculated to
result in a porewater DIC pool that was 22 atom % $^{13}$C.
Faunal uptake of added $^{13}$C was calculated by subtracting $^{13}$C attributable to its natural abundance in the
appropriate taxon (data published in Bell et al., 2017 a) from faunal isotopic signatures, and multiplying by the
quantity of organic C in each specimen. Specimens were summed for each core, and the value multiplied by 2,
to account for only half of each horizon being used for faunal extraction.

**3.    Results**

**3.1 Respiration**

Respiration rates measured in algae addition experiments varied from 0.02 mg C m$^{-2}$ h$^{-1}$ at the off vent site to
0.15 mg C m$^{-2}$ h$^{-1}$ at Middle Sister (Fig. 2).

**3.2 Bacterial uptake and PLFA suite**

In algae addition experiments, mean total bacterial uptake of C throughout the experiment was maximal at
Middle Sister and Hook Ridge (1.60 and 1.25 mg C m$^{-2}$, respectively), and minimal at the off vent site (0.51 mg

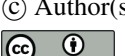



C m⁻², Fig. 3). In bicarbonate addition experiments, in which incorporation of ¹³C into bacterial PLFAs
represents chemosynthesis, bacterial incorporation of bicarbonate was maximal at the off vent site (0.077±0.034
mg C m⁻²), and was also detectable in one of the replicates at Middle Sister (0.003 mg C m⁻², close to detection
limits, so this value is treated with caution), however it was not detectable at Hook Ridge.
The PLFA suites at all sites were qualitatively similar. They were dominated by C16:0, C16:1ω7c, C18:1ω7,
and C19:0, which together constituted 58.7 ± 0.8% of total PLFAs (Fig. 4). This is relatively high compared to
36-47% in the Arabian Sea, and 41% on the Galicia Bank (Kunihiro et al., 2014). The relatively high
proportions of C16:1ω7 and C18:1ω7 are indicative of the presence of chemosynthetic and specifically sulphide
oxidising bacteria (Colaco et al., 2007). In addition C18:1ω9, which is linked to endosymbionts in vent mussels,
and C18:1ω13, which is associated with methylotrophic bacteria were also present (Colaco et al., 2007).
In both algae and bicarbonate addition experiments, ¹³C incorporation into PLFAs was dominated by C16:0,
followed by C18:1ω9 and the sulphide oxidiser indicators C16:1ω7 and C18:1ω7 (Fig 4).

**3.3 Faunal uptake**

Faunal uptake of added C was variable between A and B replicate cores in all experiments except the algae
addition at the off vent site, and bicarbonate addition at Middle Sister (Fig. 5).
In algae addition experiments faunal uptake was similar between the off vent site and one of the Hook Ridge
cores (~0.03 mg C m⁻²), while the other Hook Ridge core showed very low faunal C uptake. Considerably
greater faunal uptake (0.12 mg C m⁻²) was observed in one of the replicate cores from Middle Sister (Fig. 5).
In bicarbonate addition experiments, measurable uptake of ¹³C by fauna was observed at all sites. It was
maximal at Hook Ridge (0.02 mg C m⁻² in one replicate), and the off vent and Middle Sister sites showed
similar values (Table 2, Fig. 5).
Uptake of ¹³C in both algae and bicarbonate addition experiments was dominantly carried out by either
polychaetes, or 'mixed macrofauna' (Fig. 6). This latter category contained variously bivalves, crustaceans,
echinoderms, nematodes and foraminifera, in cases where those groups were not present in sufficient numbers
for separate reporting of their C uptake. When a group was present in sufficient quantity it was analysed
separately. As with total macrofaunal ¹³C uptake, there was considerable variability between replicate cores in
the dominant taxonomic groups. Beyond dominance by polychaetes and mixed macrofauna, one pattern to note
is contributions by meiofaunal organisms at Middle Sister, and the fact that the occurrence of bicarbonate ¹³C in



macrofaunal observed at Hook Ridge was dominantly accounted for by crustaceans, which in this case were
amphipods.
**4.   Discussion**
**4.1 Occurrence of inorganic C fixation**
The results of bicarbonate addition experiments show evidence for occurrence of benthic C-fixation at all sites,
and transfer of that C to the macrofauna, in the form of isotopic enrichment of bacterial PLFAs at the off-vent
and Middle Sister sites (Fig. 3), and of macrofauna at the Hook Ridge and Middle Sister sites (Fig. 5). The
quantities of bicarbonate $^{13}$C detected in bacterial and faunal biomass were low, and tended to be 1 to 2 orders of
magnitude smaller than equivalent values for algae addition experiments (Table 2). However, we have
confidence that the values reported are above detection limits, in that data were only used where the enrichment
of organisms or PLFAs above their natural background signatures was greater than the analytical precision of
the method. The greatest quantities of bacterial uptake were measured at the off-vent site (Fig. 3), and the
greatest quantity transferred to the fauna was measured at Hook Ridge (Fig. 5), however, due to the low values
measured and the evident patchiness of faunal communities we do not feel these differences are suitable for
further discussion.
The most striking result of the bicarbonate addition experiments was that evidence for benthic C fixation was
found at all sites, not only at the hydrothermally influenced Hook Ridge. Further, the site showing the largest
amount of incorporation of bicarbonate $^{13}$C into bacterial PLFAs was the off-vent 'control' site (Table 2, Fig. 3).
This is consistent with the occurrence of siboglinids at all sites – which host chemosynthetic endosymbionts.
However, it should be noted that the evidence for inorganic C fixation comes from PLFAs in the bulk sediment,
while isotopic signatures of siboglinids did not show significant enrichment above background values.
Therefore the occurrence of benthic C fixation is not only associated with siboglinids.
The evidence suggests therefore, that while the amount of benthic C-fixation was always low, it appears to play
a role in benthic C-cycling at a much wider range of sites and over a much larger area of the seafloor than if it
only occurred in the immediate environs of sedimented or hard substrate hydrothermal vents, methane seeps, or
organic falls (Bernardino et al., 2012). This suggestion receives recent support from the literature. Linear inverse
modelling of C-cycling at the sites in this study led Bell et al. (2017b) to suggest that chemosynthetic support
for ecosystems may have a far greater spatial extent than previously thought, extending beyond those which are
directly hydrothermally influenced. Further, similar results to those presented here have been reported in non-



hydrothermal, but methane rich sediments on the South Georgia margin, where assimilation of $^{13}$C labelled
bicarbonate into bacterial biomass, and transfer into macrofauna was also observed (Would et al., in press). In
addition, in situ observations of benthic C fixation have now also been made at mesotrophic, abyssal sites in the
eastern equatorial Pacific, which were not associated with hydrothermal or methane seep activity (Sweetman et
al. 2018). In that study incorporation of $^{13}$C labelled bicarbonate into bacterial PLFAs was observed at 2 sites
separated by 100's of kilometres, at rates similar to bacterial assimilation of phytodetritus C at the same sites.
Together with global scale modelling completed by Middelburg (2011), these studies suggest that
chemoautotrophic C fixation may be considerably more widespread than previously thought, and is an under-
studied and important aspect of the marine C-cycle.
In their study using linear inverse modelling of the benthic food web and C cycle, based on natural isotopic and
biomass data, Bell et al. (2017b) modelled a rate for chemosynthesis of 5.76-8.4 mg C m$^{-2}$ d$^{-1}$ at Hook Ridge,
and <0.006 mg C m$^{-2}$ d$^{-1}$ at the off-vent site. Thus the rates modelled at Hook Ridge are considerably higher
than Hook Ridge benthic C-fixation measured in this study, for which there was evidence (labelled PLFAs), but
a rate could not be calculated. The higher modelled rates by Bell et al. (2017 b) may be explained by the fact
that a temperature of 50°C was used for the Hook Ridge site, based on previously published conditions of the
site (Klinkhammer et al., 2001). Unfortunately, equipment was not available while at sea for measurement of
sediment temperature at the study sites, therefore all experiments, including that at Hook Ridge, were conducted
at measured bottom water temperatures of 0-1°C. It is therefore likely that higher rates of chemosynthetic
incorporation of labelled bicarbonate would have been measured at Hook Ridge if the sediment temperature
could have been measured at the time of sampling, and used as the experimental condition. It is also probable
that measurable rates could have been detected at Hook Ridge had more samples been available for replicate
analyses.
The maximal rate of benthic C-fixation measured in this study was 0.050 mg C m$^{-2}$ d$^{-1}$ , which occurred in one
core at the off-vent site. This remains considerably lower than the 0.24-1.02 C m$^{-2}$ d$^{-1}$ measured by Molari et al.
(2013, rates calculated in Sweetman et al., 2018) at depths ranging between 1207-4381 m on the Iberian margin
and in the Mediterranean, and the 1.29 C m$^{-2}$ d$^{-1}$ measured by Sweetman et al. (2018) at ~4100 m depth in the
Clarion Clipperton Zone. The Bransfield Strait sites in this study were shallower, had higher concentrations of
sedimentary organic C, and slightly lower bottom water temperatures than either of the previous studies cited.
The very low temperatures at which experiments were conducted (1°C at Hook Ridge and 0°C at the off vent
site) is likely to have contributed to the slow measured rates of benthic C-fixation. Another factor which may



influence benthic C-fixation is the annual flux of photosynthetic C from the surface (Molari et al., 2013; Bell et
al., 2017a). The annual flux of POC to the sediments in the Bransfield Strait is greater than in the Clarion
Clipperton Zone, and probably than in the Mediterranean as well (Masque et al., 2002; Sweetman et al., 2017),
therefore this may be an additional driver behind the low benthic C-fixation rates observed. In addition, archaeal
abundance has been show to correlate with dark C-fixation, and addition of labile organic material has been
shown to increase inorganic C fixation rates, perhaps through a combination of heterotrophy and mixotrophy
(Molari et al., 2013). Therefore the factors governing benthic C-fixation rates require investigation. In addition,
the pathways (i.e. autotrophic C fixation versus anapleurotic C fixation by heterotrophs, Wegener et al., 2012),
energy sources (e.g. sulphide, methane) and organisms responsible for benthic inorganic C fixation have not
been identified, and warrant further study.
4.2 Carbon uptake by macrofauna
Uptake of added C by fauna in isotope tracer experiments usually shows a degree of spatial patchiness (e.g.
Woulds et al., 2007), but this seems to have been particularly marked in the Bransfield Strait, mainly at those
sites with hydrothermal influence. This is consistent with the patchiness of *Sclerolinum contortum* in replicate
cores at Hook Ridge (Bell et al. 2016a). At both Hook Ridge and Middle Sister there was a very marked
difference in faunal uptake of algal C between the A and B replicate cores in algae addition experiments (Fig.
5). On the Pakistan margin, Woulds et al. (2007) noted that the average relative standard deviation in faunal C
uptake between A and B replicate cores was 42%, whereas for all Bransfield Strait sites this value was 92%.
This is likely to be due to difference in the biomass of fauna present in each core, and such marked small scale
patchiness in faunal communities has been noted previously as a particular feature of SHVs (Levin et al., 2009;
Bernardino et al., 2012).
Faunal uptake of added C appeared to be greatest at Middle Sister in algae addition experiments, and at Hook
Ridge in bicarbonate addition experiments, however the variation between replicate cores limits conclusions that
can be drawn. Previous isotope tracing experiments have noted correlations between biomass of organisms and
taxa and the amount of C they take up (e.g. Woulds et al., 2007). Further, there was no systematic variation in
biomass-specific C uptake (0.026-0.13 ug C uptake / mg C biomass) between sites, therefore the patterns
observed here in faunal C uptake are likely to result from variation in biomass present in each experimental
core.



Similarly, the identities of fauna responsible for [13]C uptake was rather variable between replicate cores (Fig. 6),
and this is also likely to have been driven by variation in the macrofaunal community present in each core. The
prevalence and variable importance of the 'mixed macrofauna' category indicates that in some cases a fairly
diverse assemblage was engaged in C uptake and processing.
Previous studies have suggested that SHVs tend to exhibit relatively high biomass macrofaunal communities,
sustained by the additional food source provided by chemosynthesis (Bernadino et al., 2012), and this leads to
an expectation that the macrofauna may be particularly active in processing of organic C in the sediment, in line
with other food rich environments such as estuaries and fjords (Moodley et al., 2000; 2005; Witte et al., 2003a).
This was not the case in the algae addition experiments however, with faunal uptake accounting for only 0.05-
2.2 % of total biological [13]C processing (Fig. 7). This is lower than the role of faunal C uptake in overall C
processing seen at deep, organic carbon poor sites such as at 2170 m depth off NW Spain (Moodley et al.,
2002), or at 1552 m depth in the Eastern Mediterranean (Moodley et al., 2005). However, such sites tend to have
lower OC concentrations and lower macrofaunal biomass (Woulds et al., 2016) than was observed in the
Bransfield Strait, therefore the unusually small role of macrofaunal in C uptake in the Bransfield Strait may be
due to low temperatures. Another possible explanation for the rather small amount of macrofaunal C uptake at
the Hook Ridge site may be that the macrofaunal community, which was composed almost entirely of non vent-
obligate, ambient Southern Ocean taxa (Bell et al., 2016a), had reduced levels of function due to the stress
imposed by living at a site influenced by hydrothermal fluid. Thus, the toxicity and relatively high temperature
of their environment (compared to non-hydrothermal Southern Ocean benthic settings) may have resulted in
reduced C uptake activity. Therefore, macrofaunal biomass and C processing activity were limited by a
hydrothermal flux that was sufficient to impact ambient background taxa, but insufficient to sustain a high
biomass, vent endemic macrofaunal community as seen in other SHVs (Bell et al., 2016 a).
Siboglinid polychaetes, known to host chemosynthetic endosymbionts, were present at all study sites (Bell et al.,
2016 a), however they were not found to make a significant contribution to uptake of added [13]C. This is to be
expected in the algae addition experiments, as the siboglinids were not expected to have direct access to algal C
(although they may have been able to access any which was released as DOC). Most specimens recovered from
biocarbonate addition experiments also showed $\delta$[13]C values indistinguishable from their natural signature, with
one exception at the Middle Sister site, which was enriched compared to the natural signature by 3.2 ‰. The
fact that siboglinids did not have a major role in C fixation and cycling in our experiments may have been partly
due to their low abundances in experiment cores compared to patches where they were maximally abundant



(Bell et al., 2016a) Nonetheless, our findings show a much reduced role for siboglinids compared to suggestions
made in previous publications. Aquilina et al. (2014) suggested that Siboglinum at Hook Ridge may be
sufficiently abundant to be conduits for a quantitatively significant flux of dissolved iron out of the sediment,
and Bell et al. (2017 b) found that they may be a key taxon facilitating input of chemosynthetic C into the food
web. In agreement with the point made by Bell et al. (2016a), the spatial distribution of siboglinids is extremely
patchy, and thus their role in benthic biogeochemical processes is spatially heterogeneous (Bell et al., 2017a, b).
4.3 Carbon processing and SHVs as biogeochemical hotspots
Respiration rates measured in the algae addition experiments were maximal at Middle Sister, and minimal at the
off-vent site (Fig. 2). Temperature is often recognised as a dominant control on benthic respiration rates (e.g.
Moodley et al., 2005; Woulds et al., 2009), however these experiments were all conducted within 1°C of each
other, so temperature is unlikely to have driven differences in respiration rates. Instead, the differences between
sites may have been driven by differences in bacterial biomass (Table 1), which was maximal at Middle Sister
and minimal at the off-vent site. The bacteria are often found to account for a large majority of benthic
community biomass, and are thus usually assumed to be responsible for the majority of benthic community
respiration (e.g. Heip et al., 2001). The measured respiration rates were similar to those measured at 2170 m on
the NW margin of Spain (Moodley et al., 2002), and on the Porcupine Abyssal Plain (Witte et al., 2003b), both
of which were considerably deeper, and had lower sediment organic C concentrations, but higher bacteria
biomass (Woulds et al., 2016). They were also lower than respiration rates measured at similar depths in the
Eastern Mediterranean (Moodley et al., 2005), and Arabian Sea (Woulds et al., 2009). These sites showed
similar bacteria biomass to the Bransfield Strait, but were all considerably warmer (7-14°C, Woulds et al.,
2016), therefore the low ambient temperatures of the Southern Ocean did appear to reduce respiration rates
overall.
It has been suggested that reducing benthic environments are often hotspots of faunal biomass and
biogeochemical cycling due to the increased availability of labile food sources supplied by chemosynthesis
(Bernardino et al., 2012), and thus high biomass benthic communities. In this study, the hydrothermally active
site Hook Ridge showed rates of respiration and bacterial uptake of algal C that were intermediate between the
two non-hydrothermally active sites (Figs. 2, 3). Further, while comparison between sites is limited by very
marked faunal patchiness, the amount of faunal uptake of algal $^{13}$C at Hook Ridge was similar to that at the off-
vent control site, while that at Middle Sister was, in one replicate, considerably greater (Fig. 5). Therefore this
suggests that SHVs are not necessarily biogeochemical cycling hotspots, as in algae addition experiments the



overall amount of added C processed by the benthic community was not greater than that observed at non-
hydrothermal sites (Fig. 8). In line with this, biological processing of added C in the algae addition experiments
did not show a significant role for faunal C uptake as we hypothesised, but was instead dominated by
respiration, as is typically observed at relatively deep, cold sites (Woulds et al., 2009). The Middle Sister site
showed the greatest amount of biological processing of added algal C, which was probably attributable to it
having the greatest bacterial biomass and organic carbon concentrations, and the fact that the macrofaunal
community, composed mostly of ambient Southern Ocean taxa, will have been functioning without the stress
imposed by hydrothermal fluid.
**5. Conclusions**
The dominant fate of photosynthetic C was respiration in common with other deeper and more food limited
sites. The rates of respiration and C uptake by both macrofaunal and bacteria that we measured were
comparatively low, and this is attributable to the low temperature of the experiments, and the toxicity and
thermal stress caused by hydrothermal fluid. Therefore the hydrothermal site (Hook Ridge) in this study was not
the hotspot of C-cycling that we hypothesised it would be.
Benthic fixation of inorganic was observed at all sites, and quantified at 2 out of 3 sites. While the rates were
low compared to other similar experiments, the fact that the greatest amount of benthic C fixation occurred at
the off vent site suggests that benthic C fixation may not be restricted to hydrothermal and other reducing
settings. We suggest that it could be an important aspect of the marine C-cycle, and warrants further study.
**Data Availability**
Data sets can be found at DOIxxxx
**Author Contributions**
Experiments were conducted by C. Woulds and A. Glover. All authors contributed to analysis of samples, and
commented on the manuscript.
**Acknowledgements**
This work was funded by Antarctic Science Ltd., and NERC (grant NE/J013307/1). The authors would like to
thank Prof. Paul Tyler, as well as the officers and crew of RRS James Cook, and the on-board scientific party on
cruise JC 55. We would also like to thank Elisa Neame for assistance with extracting macrofauna.



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





| Site | Lat. | Long. | Depth (m) | Temperature | Sediment wt%Corg in 0-1 cm horizon | Macrofaunal Biomass (mg C m$^{-2}$) | Bacterial Biomass (mg C m$^{-2}$) |
|---|---|---|---|---|---|---|---|
| Off-Vent | 62.3842 S | 57.2440 W | 1150 | 0 | 1.35 | 1091 | 314±145 |
| Hook Ridge | 62.1924 S | 57.2783 W | 1054 | 1 | 0.97 | 318 | 451±21 |
| Middle Sister | 62.6552 S | 59.0502 W | 1311 | 0 | 1.40 | 374 | 575±394 |

**Table 1. Site characteristics, all except bacterial biomass are from Bell et al. (2016).**




| Site | Treatment and Replicate | Amount Respired (mg C m$^{-2}$) | Respiration Rate (mg C m$^{-2}$ h$^{-1}$) | Bacterial Uptake (mg C m$^{-2}$) | Macrofaunal Uptake (mg C m$^{-2}$) |
|------|------------------------|--------|----------------|----------|------------|
| Off-Vent | Algae A | 1.23 | 0.025 | 0.25 | 0.027 |
| Off-Vent | Algae B | 0.75 | 0.015 | 0.77 | 0.034 |
| Off-Vent | Bicarbonate A | N/A | N/A | 0.053 | 0.0009 |
| Off-Vent | Bicarbonate B | N/A | N/A | 0.102 | low |
| Hook Ridge | Algae A | 4.97 | 0.087 | n.d. | 0.033 |
| Hook Ridge | Algae B | 4.06 | 0.071 | 1.25 | 0.003 |
| Hook Ridge | Bicarbonate A | N/A | N/A | n.d. | 0.021 |
| Hook Ridge | Bicarbonate B | N/A | N/A | low | low |
| Middle Sister | Algae A | 7.16 | 0.13 | 1.91 | 0.004 |
| Middle Sister | Algae B | 8.37 | 0.15 | 1.30 | 0.12 |
| Middle Sister | Bicarbonate A | N/A | N/A | 0.00 | 0.003 |
| Middle Sister | Bicarbonate B | N/A | N/A | 0.003* | 0.003 |

**Table 2. Amount of C in pools at experiment end, and respiration rates (algae addition experiments only). N/A indicates**
**that it was not appropriate to measure respiration in bicarbonate addition experiments, n.d. indicates no data due to**
**missing sample, and 'low' indicates unmeasurably low value. The value marked \* indicates detectable bacterial $^{13}$C**
**uptake, but very close to detection limits, so value to be treated with caution.**



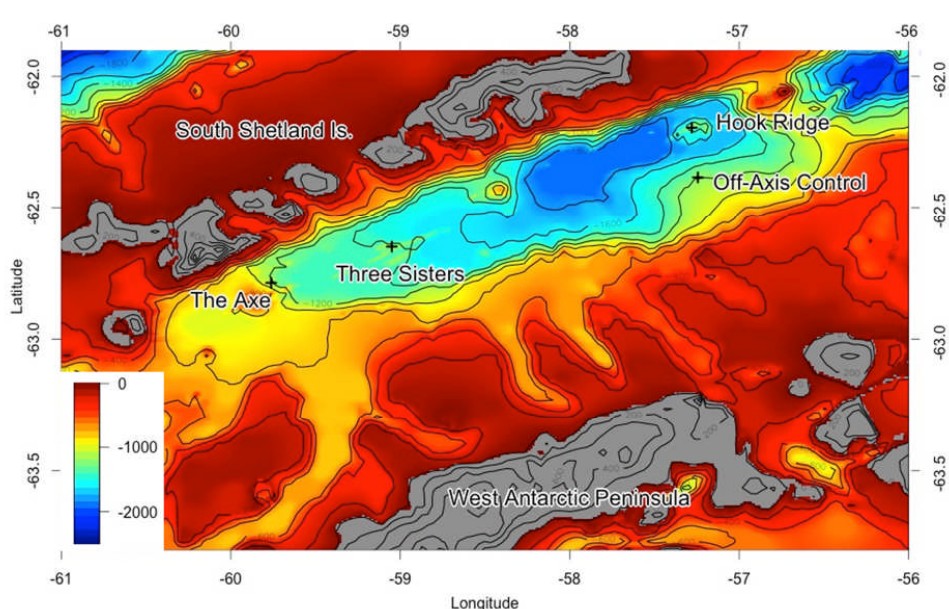


**Figure 1. Map of study sites, adapted from Bell et al. 2016 a. The Off Vent site is marked 'Off-Axis Control', and the**

**Middle Sister site is located where 'Three Sisters' is marked.**






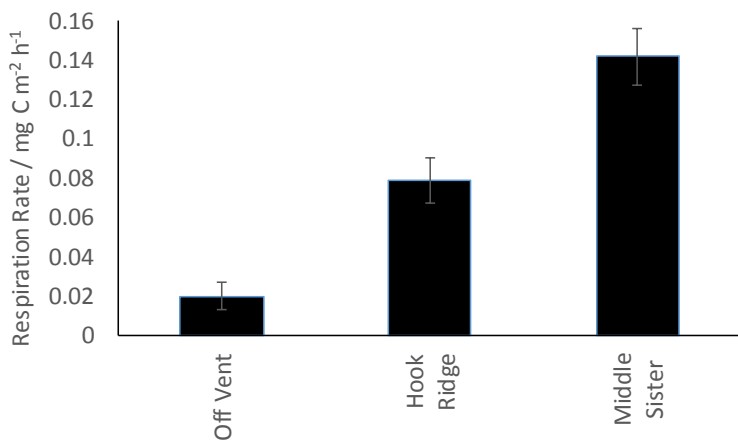


**Figure 2. Respiration rates measured in algae addition experiments. Error bars are ± 1 standard deviation.**



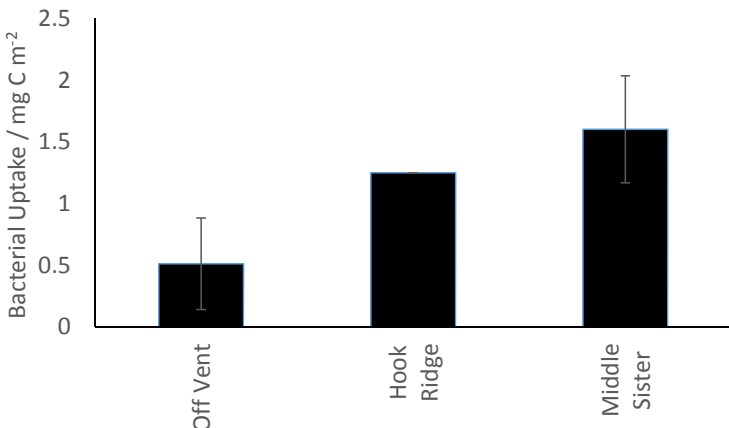


A

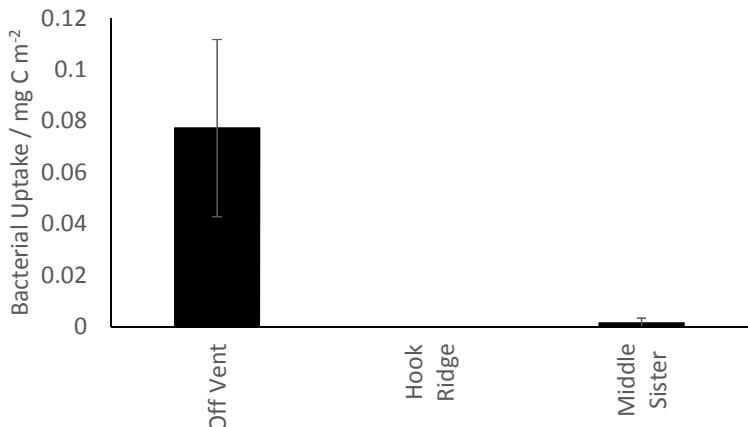


B
**Figure 3. Bacterial uptake measured in A) algae addition experiments; B) bicarbonate addition experiments. Error**
**bars are ± 1 standard deviation. Error bars are not plotted for Hook Ridge because replicate samples were not**
**available.**






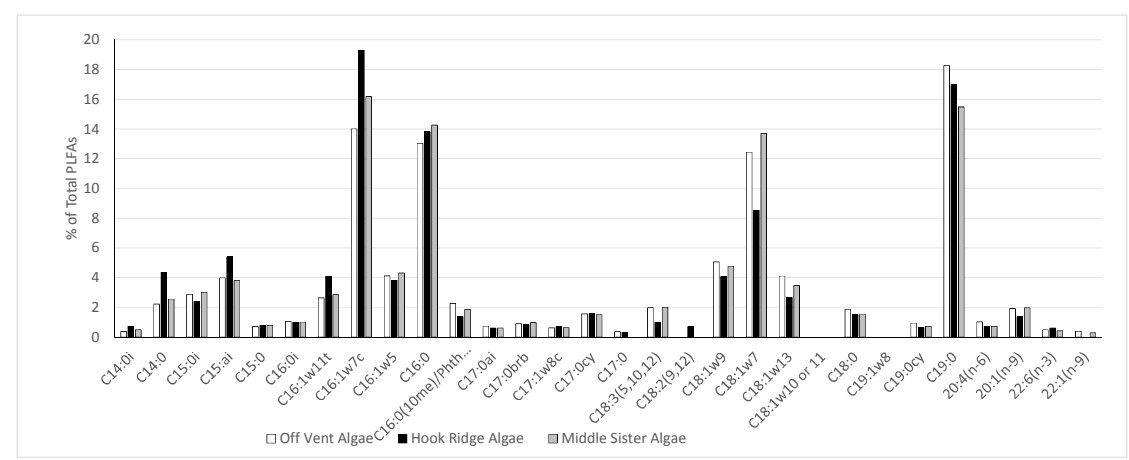


A

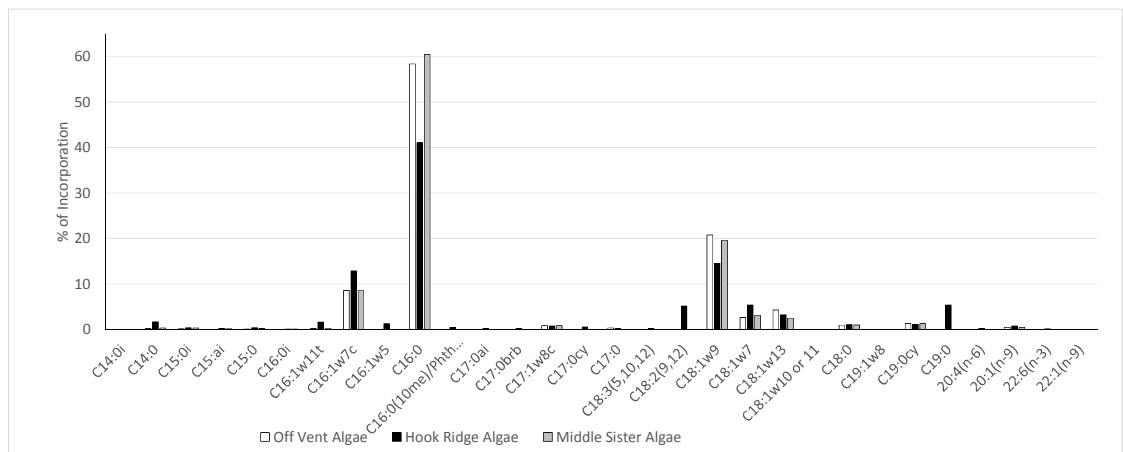


B





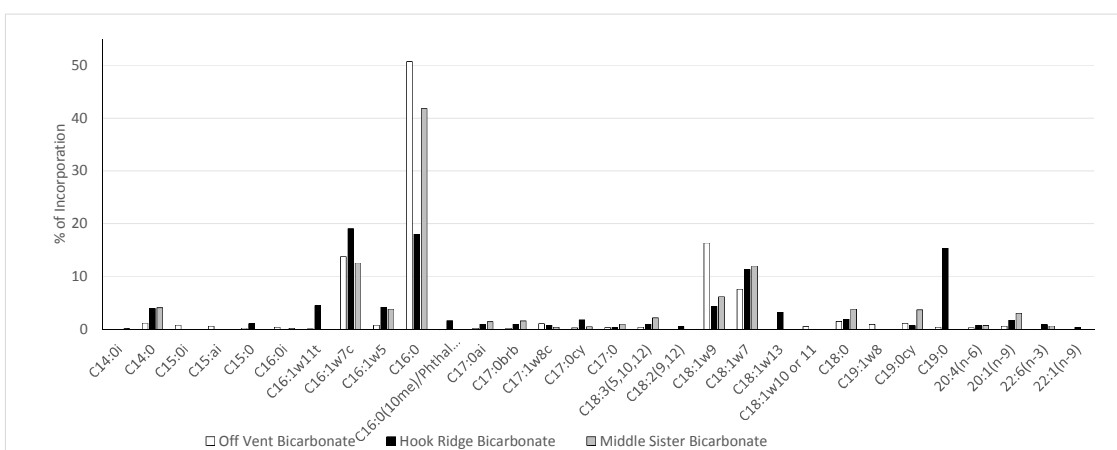

C

**Figure 4. Example PLFA suites – each data series is from one sample, as opposed to being an average across two replicates. A) PLFA suite as % of total PLFAs in algae addition experiments (figure for bicarbonate addition experiments very similar and not shown), B) Composition of $^{13}$C uptake into PLFAs in algae addition experiments, and C) Composition of $^{13}$C uptake into PLFAs in bicarbonate addition experiments.**





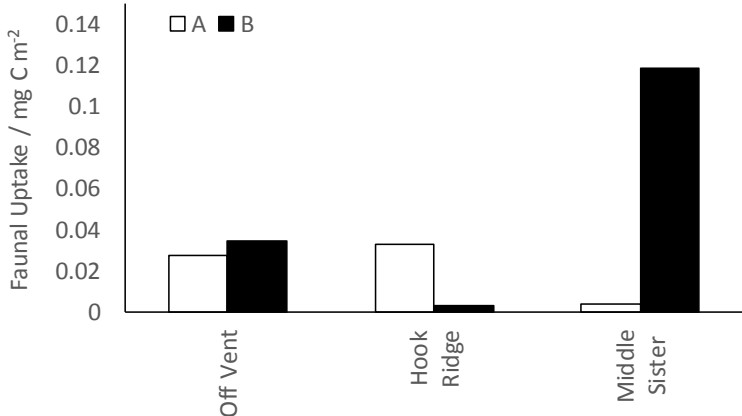


A

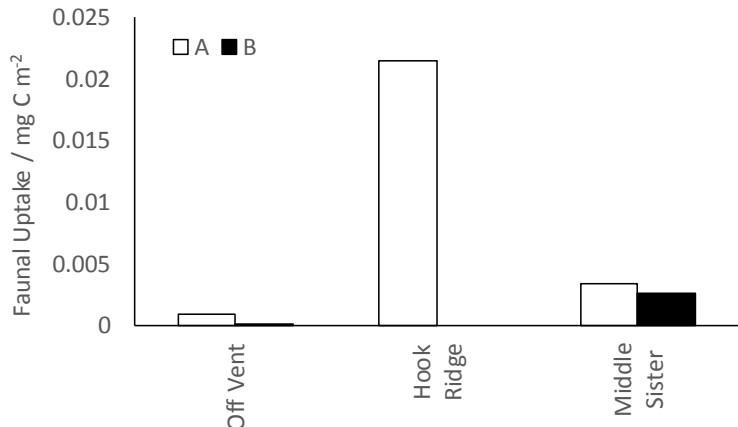


B
**Figure 5. Faunal uptake in A) algae addition experiments, and B) bicarbonate addition experiments. A and B refer to**
**the two replicate cores in each experiment.**



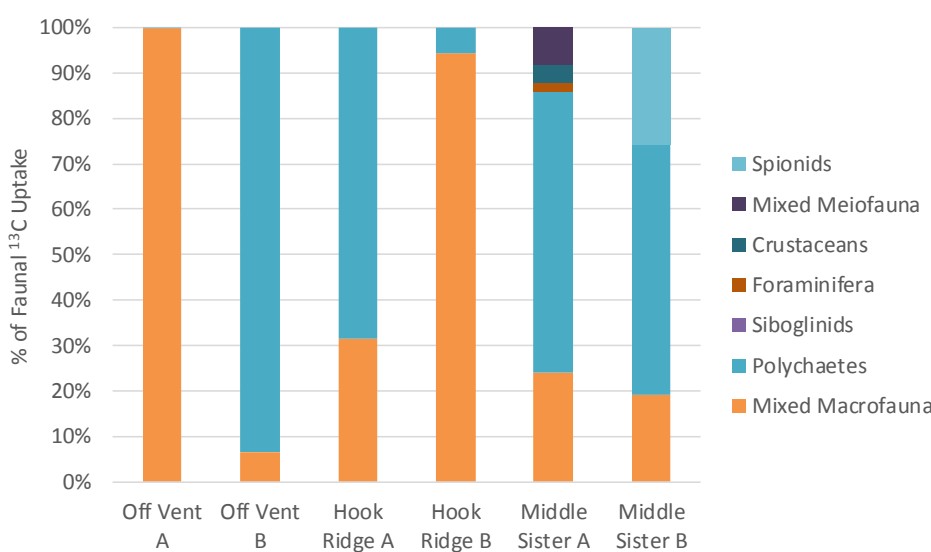


A

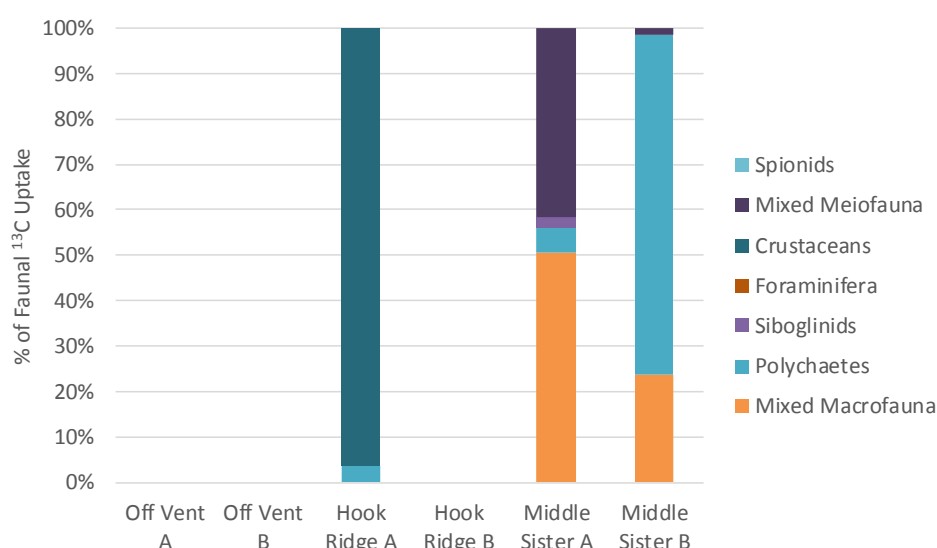


B
**Figure 6. Distribution of C uptake amongst taxonomic groups in A) algae addition experiments, and B) bicarbonate**
**addition experiments.**





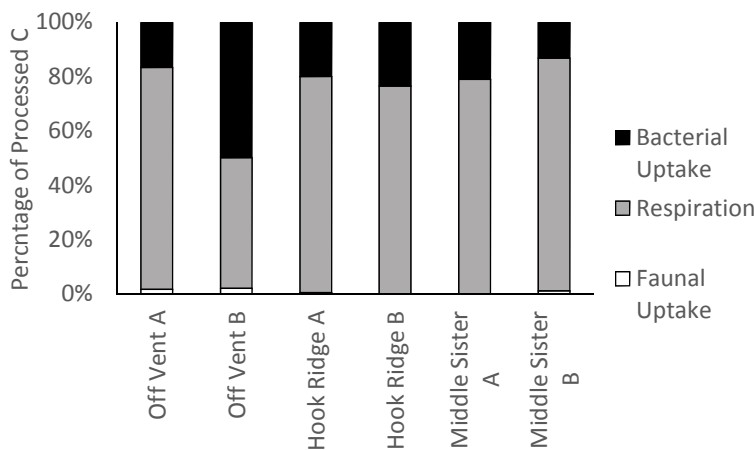


A

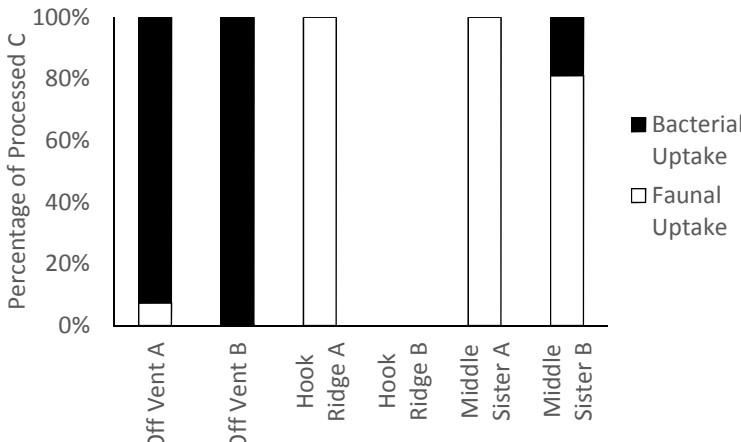


B
**Figure 7. Distribution of biologically processed C between processes for A) algae addition experiments, and B)**
**bicarbonate addition experiments.**




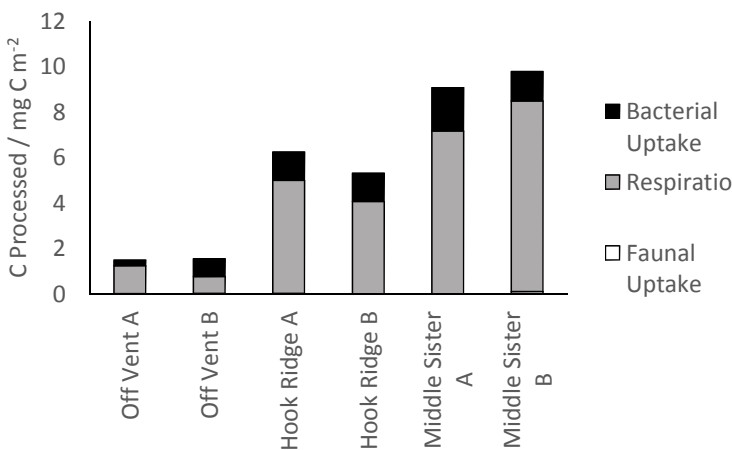


A

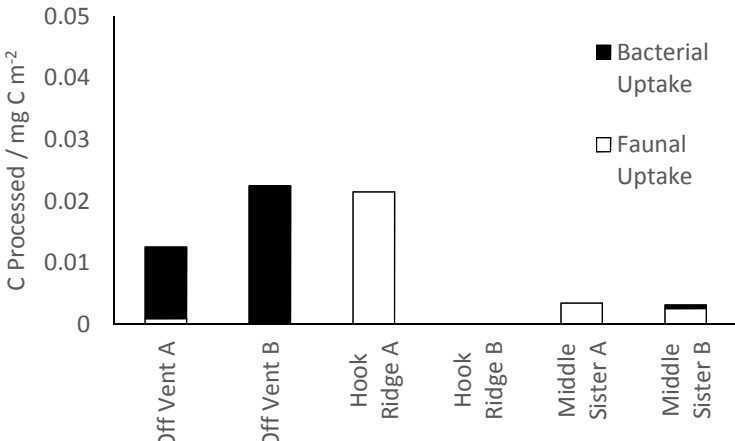


B
**Figure 8. Total biological C processing during A) algae addition experiments, B) bicarbonate addition experiments.**