# Peer review of "and background sediments in the Bransfield Strait,"

_Biogeosciences, 2019_

## Referee Comment (RC1) · Anonymous Referee #1 · 26 Jun 2019

**Review of Woulds et al. 2019. Benthic C fixation and cycling in diffuse hydrothermal and**

**background sediments in the Bransfield Strait, Antarctica. DOI: 10.5195/bg-2019-198.**

**General Comments**

The manuscript by Woulds et al. describes a series of stable-isotope pulse chase experiments conducted at three sites, associated with diffuse venting of hydrothermal fluids in Antarctica. The paper is novel and describes some elegant experiments which seek to disentangle the role of chemosynthetic pathways in the benthic carbon cycle at these sites.

The paper provides compares benthic fixation of $^{13}$C-labelled bicarbonate and processing of

13C-labelled phytodetritus, to determine the relative role that chemosynthetic pathways play within the benthic carbon cycle of diffuse hydrothermal vents and encompasses measurements of both bacterial and macrobenthic $^{13}$C-uptake. This provides a very exciting study, which allows the relative roles of the bacteria and fauna to be compared between the study sites. This study is ambitious and provides a valuable contribution to our understanding of the biogeochemistry of deep-sea sediments. However, it also has a number of notable flaws that need to be addressed prior to publication. These are outlined below:

**Main points requiring revision.**

1. Given that you processed only half a 10 cm core for either bacterial or faunal analysis, care needs to be taken in the interpretation and extrapolation of the data.

Each replicate of the experiment sampled only 0.0039 $m^2$ of the seafloor for either the faunal or bacterial community. Subsequently the data is scaled to units.$m^{-2}$ which is an area ~256 times larger than the area sampled. Like many colleagues in the deep-sea and marine ecology communities, I have made very similar decisions with some of my own papers. However, I wonder whether this is really an appropriate standardization to make. Scaling our data like this inevitably propagates errors from single core incubations up to geographically relevant macro- and meso-scales. I

would ask the authors to consider standardizing their data to a smaller areal size (such as $cm^2$ or 10 $cm^2$). This would provide a more honest description of the results.

2. Error Bars reporting standard deviations are plotted on Figures 2 and 3, yet only two replicate cores where incubated at each study site. Based upon a sample size of *n*=2

it does not make sense to calculate a mean or standard deviation, as the mean will always be halfway between the two values. Please remove reference to the standard deviations as a measure of variation within the text and revise Figures 2 and 3 to show the individual values for each replicate (as you have done in Figure 5). In terms of future experimental design, $n = 2$ is not really an adequate sample size to allow assessment of differences between sites (for details refer to Sokal and Rolff, 1994, or

Underwood, 1997). The novelty of these experiments as an observational study of carbon cycling in a poorly explored region of the oceans, however, warrants their publication.

I also have a number of minor comments which I would ask the authors to address prior to publication.

**Minor Comments**

***Title***

Page 1 Line 1: Should read "Benthic carbon fixation…."

***Abstract***

Page 1 Line 14-15: "There are no previous direct…" This sentence is not required. Please delete the sentence.

Page 1 Lines 15-16: Remove paragraph break.

Page 1 Lines 21-22: Remove paragraph break.

Page 1 Line 22: Revise to: 'Fixation of inorganic C into bacterial biomass was observed in all cores/sites.' Please revise as suggested.

***Introduction***

Page 3 Line 30 – Page 4 Line 85: Throughout the introduction there are many uses of

'therefore' and 'however'. 90 % of the time these words are superfluous. Please revise the introduction to make less use of them.

Page 3 Line 32. Split this into two sentences. '…dissolved sulphides and methane. This supports microbes that combine…'

Page 4 Line 63-64: Do you have any supporting literature that can be cited to support this sentence.

Page 4 Line 72: 'On the contrary however' please revise, this is not well phrased.

Page 4 Line 77: Delete sub-heading

Page 4 Line 80: Hypotheses should be 'tested' not 'addressed'

***Methods***

Page 5 Line 108-119: This provides a brief summary of the experimental methods. Please refer to an alternative source as (following…) where a more detailed description of the method can be found.

Page 5 Line 111-112: *Chlorella spp.* phytodetritus would not be representative of the algal material processed in Antarctic systems. A diatom would have been a more appropriate choice of $^{13}$C-labelled substrate.

Page 5 Lines 119-119: Half a core seems to be a very small volume of sediment for conducting macrobenthic analysis. Given that the size range of macrobenthic fauna is variable, and species are mobile, is this sample volume appropriate? I get the impression that you may be missing something significant by only focusing on half a core for the bacterial and macrobenthic communities.

Page 6 Lines 151-158: A lot of potential data has been discarded from the PLFAs by just focusing on four 'bacteria-specific' fatty acids. It would be interesting to see the full profiles, particularly as the 13C-labelled bicarbonate treatment may reveal some insight into which

PLFAs might be good indicators of microbial carbon fixation.

***Results***

As previously mentioned, I am not content with the use of standard deviations to describe variation in the data. Where n = 2, you cannot reliably calculate means or standard deviations.

Page 7 Line 168: 'In *the* algae addition experiments…' Please revise.

Page 8 Lines 174-181: I think you could potentially offer more insight into the microbial processes by considering a wider range of PLFAs for each site. Which PLFA groups showed greatest label uptake?

Page 8 Line 175: Normally C19:0 is used as a standard in the PLFA analysis which may explain why it is found in higher concentrations.

Page 8 Line 183: Please revise to 'Faunal uptake of added C differed between the two replicate cores in all experiments…'

Page 8 Line 191-192 and Figure 6: Given the small sample size, I am not convinced that a community level analysis of faunal feeding responses is appropriate. Differences in faunal uptake are likely to be driven by spatial variability, with common taxa such as polychaetes heavily overrepresented. This leads to the 'mixed macrofauna' category essentially consisting of everything except polychaetes.

Page 8 Line 191 – Page 9 Line 199: In light of the small sample size please don't refer to dominance either in terms of faunal abundance or feeding responses. It would be more appropriate to discuss simply which groups were more/less abundant and exhibited greater/weaker uptake of the $^{13}$C-label.

Page 8 Line 196- Page 9 Line 199: This last sentence is confusing, please revise and clarify.

***Discussion***

There is frequent use of 'therefore' and 'however', please remove these where possible.

Page 9 Lines 202-212: This paragraph is a description of the results. Please revise to contextualize your findings.

Page 9 Lines 220-222: Long sentence, requires broken up. Please revise.

Page 9 Lines 223 Page 10 Line 228: Please revise along the lines of "This is supported by a recent modelling study which suggested that…. (Bell et al 2017b). Similar results have also been reported from the methane-rich non-hydrothermal sediments… (Woulds et al., in press).

Page 10 Line 242-248: I am afraid that this is a major flaw in the overall paper. Given that temperature is critical to microbial metabolism, the current paper is likely to seriously underestimate the level of carbon fixation. This needs to be made clearer earlier in the paper.

Page 11 Lines 273-275: Based on two replicates, it would only be possible to discuss the magnitude of the differences and perhaps compare these between sites. Remove reference to standard deviations from the discussion.

Page 12 Line 285: Delete 'rather'

Page 12 Line 298: Revise to '…Branfield Strait. Therefore…'

Page 12 Lines 298-299: Here you are discussing the effects of temperature on metabolic rates. Here you should consider the impacts of rate limitation and do a quick literature search. There is quite a large body of literature on this topic.

Page 12 Line 302: Delete 'Thus'

Page 13 Line 324-325: Based on the $Q_{10}$ effect, metabolic activity increases logarithmically with temperature. As such, a change of $1^{o}C$ may be more significant than you assume. I

think this may require further explanation.

Page 13 Line 339: Delete 'and thus high biomass benthic communities.'

Page 13 Line 341: Delete 'Further' and replace 'while' with 'Whilst'

Page 13 Line 341-342: The comparison between sites was limited by the size of each sample (half a core), and lack of replication (n = 2). Your experimental design does not allow you to make any inferences on faunal patchiness.

Page 14 Line 347: You cannot use the term 'significant' as this implies the use of inferential statistical tests. Please revise.

Page 14 Line 354: Replace 'dominant' with 'main'

Page 14 Line 357-358: 'Therefore the hydrothermal site (Hook Ridge) in this study was not the hotspot of C-cycling that we hypothothesised it would be.' You need to define what is meant here by a 'hotspot of C-cycling.' Is this referring to chemosynthetic carbon fixation?

Page 14 Line 358-359: Delete paragraph break.

Page 14 Line 362: Delete the final sentence.

References

Sokal, R.R., Rolff, F.J. (1994). Biometry: The Principles and Practices of Statistics in Biological

Research [880 pp.]. New York, USA: W.H. Freeman.

Underwood, A. J. (1997). Experiments in ecology: Their logical design and interpretation using analysis of variance (524 pp). Cambridge, UK: Cambridge University Press.

---

## Referee Comment (RC2) · Anonymous Referee #2 · 19 Jul 2019

The authors evaluated importance of two C sources to benthic community at sedimented hydrothermal vents; one as photosynthetic C and the others as chemosynthetic C. They conducted 13C-incubation experiments onboard using cores obtained from 3 different stations from the active venting site to the reference (normal sediment) site. The results somewhat contradict to the author's (and readers) expectation, that the chemosynthetic OM production was extremely low at the active venting sites but higher at non-vent site. Vent characteristic polychaetes Siboglinid including symbionts also did not exhibit any sign of OM production thorough 13C-bicarbonate.

As the authors also mentioned in the text, they carried out the experiment at bottom

water temperature, while the sedimented venting sites should have high temperature at the subsurface sediments. Furthermore, more importantly, chemoautotrophic activities at the sedimented venting site must be supported by the reducing compounds contained in the venting fluid. However, the incubation experiments did not have this supply from the deeper part of the sediments. I indeed agree that it is extremely difficult to simulate the venting fluid from the bottom in the laboratory, but without that supply, the measured chemoautotrophy should have decreased dramatically because chemoautotrophic C production at the vent is supported by both oxic seawater and reducing venting (or seepage) fluid. I therefore think the measured C production rates using bicarbonate did not reflect that of in situ.

I think the manuscript should focus only on the importance of phytodetritus C consumption at normal and sedimented hydrothermal vent sites. Even though there were only 2 replicate cores at each site (and have high variations among a site, making it difficult for proper interpretations), it may be worth to report because there is no such experiment performed at hydrothermal vent area.

I noted some specific comments below.

Line 57 to 62 It is better to mention about the time scale, because C uptake by fauna will also be respired into CO2 in longer time scale.

Line 93 Middle sister is not described in Fig 1 (Three Sisters is described). Off-vent is also not described in Fig 1, but Off-Axis control is described. Please be consistent through the text and the figures.

Table 1 The authors listed the water temperature of each site, but do you have a data for characterizing each site in terms of venting activities, such as heat flow value, H+ or CH4 or Cl concentration of pore water? Can you list up some from Aquilina et al. 2013??

Line 115 (If the authors decided not to delete Chemoautotrophic C production results)

To give better idea how much 13C- and 15N were dosed into existing DIC or ammonium, it is needed to indicate them uM. In the line 158, the authors mentioned that the added 13C-DIC account for 22%, but this must differ between sites because venting fluid contains high DIC (5-100mM) than bottom water ($\sim$2.1mM).

Line 151 It is totally unclear which depths did authors use for each analysis. For PUFA, all sediment layers were used or not? For faunal, the authors examined 10 cm or deeper?

Line 173 More specifically, how much 13C-labeling was determined as detection limits considering natural variations in d13C values? This must be written in M&M.

Line 188 Again, how much 13C-enrichments were regarded as 13C uptake? "Measurable uptake" sounds like even 1 per mil of d13C differences from background are regarded as uptake.

Line 206 Please describe this for more detail. Not only analytical precision, but also variations in background samples in replicate (if available) must be considered, which sometimes shows $\sim$5 per mil of variation.

Line 213 If you measured 13C of PLFAs in different layers, it is worth to put vertical trends in the graphs.

Lines 250 and 252 Probably, "mg" is missing.

Line 273 It is odd that describing "standard deviation" on samples with 2 replicates.

Figure 7 (If the authors decided not to delete Chemoautotrophic C production results) It is a bit confusing to show these graphs together; one with "respiration" but one without. I understand that the respiration of B cannot be measured because of 13C-DIC addition, but you need to mention that clearly in the caption.

Line 325 The differences in microbial biomass were less than twice, while those in respiration rates differed $\sim$7 times. So, the microbial biomass is not the only reason.
The authors need to discuss about these fact more carefully.

---

## Referee Comment (RC3) · Anonymous Referee #3 · 22 Jul 2019

General comments: This manuscript investigates the uptake of 13C via two substrate types 1) freeze dried algal cells and 2) bicarbonate into ∼1000 m depth sediments with two sites being sedimented hydrothermal vents with different levels of diffuse venting and an off-vent control from a similar depth. The study finds that respiration was the dominant pathway for processing of algal material and that chemosynthesis was non-zero in all sites for both bacteria and fauna, but higher in the non-vent control. Chemosynthesis was subsequently confirmed through uptake of 13C into PLFAs for all sites, with non-vent uptake largely mirroring the patterns found for % incorporation of 13C. The authors thoroughly acknowledge that due to the limited replication and temperature differences between in situ and lab, these rates represent a confirmation of non-zero rates and are not sufficient to provide further comparisons between sites for fixation of C. The authors have provided a study, that despite the acknowledged limitations, demonstrates the dominate pathway for heterotrophy for particulate OM and confirm chemosynthesis (both background and HSV) without overreaching in their interpretations. I find the manuscript clearly written, but overly concise in the methodology. The results presented adequately support the authors interpretations, with context presented for the limitations present in the study. Specific comments: LN 83: "inorganic substrates" to bicarbonate (H13CO3-) LN 103: A brief description here of the background macrofauna would be appropriate. LN 112: Product number for your labeled algal material is needed, CIL does not appear to sell a marine algal detritus that I could find by searching their site. Be careful with that description too, as it implies a bit of possible reworking given that you are working at 1000 m depth. If the material is the dual labeled lyophilized algal cells then it is really fresh algal material for application at a relatively deep site; you should discuss this if it is the case. An estimate of what portion of the annual flux this application represents would be appropriate to give more context to the amount of material be applied. LN113: Context for the relative amount of application versus what is already there and available would be useful so the reader can gauge how large the applications are versus in situ backgrounds for C and N. LN 116: Describe the sampling intervals, this will help to indicate how many measurement points your rates are determined off of. LN 206: Provide a range of PLFA or organic ‰ 13C enrichments to support this claim. It will be more convincing to readers when presented in that manner. LN 212: I appreciate the candid nature of this statement, it shows a realistic interpretation of the data given the limited replication built into the study. LN 216: Provide a reference about the chemosynthetic endosymbionts, also an indication as to the nature of the symbionts, methane oxidizers or sulfur oxidizers would be appropriate. LN 235: "important aspect" Is this because it is minor, but potentially widespread? Vague as written. LN 244: So, this study likely represents minimum rates for chemosynthesis. The authors should phrase it that way and provide context of what the addition represented in comparison to normally available substrates. Better

to focus on what your study has actually shown than to speculate that rates would have been higher if in situ temps were maintained. LN 250 & 252: 0.24-1.02 and 1.29 both need mg in front of C m-2 d-1, respectively LN 263: Would it be worth trying to isolate polar lipids from archaeal components given their slow metabolism and the relatively short time frame of this study? LN 268-277: Thank you for addressing the variability observed during tracer studies relying on bacterial mediation of a substrate! Is it worth talking about reasons for potential hotspots for both heterotrophy and chemosynthetic processes that are occurring in this system? I would expect variations in vent flows and sporadic availability of resources to give rise to a community that readily adapts to changing conditions. LN 294: Provide percentages from the other studies here so the reader can directly compare these studies. LN 316: Does the time period involved in this incubation matter here? Transfer into symbiont and then into tube worm may take a bit more time and require a stronger signal to show up as the tracer is sequentially diluted through the two carbon pools? Figures: Figure 1: state that depth is in meters in figure caption. Figure 2 & 3: remove blue outline on bars. Considering the low uptake rates, consider converting into $\mu$g to limit the decimal places. But, you are consistent throughout currently. Figure 4: Format the letters for the figures into the actual graphs, hard to interpret as laid out presently. Also resulted in the splitting of the figure between page 24 and 25. Both substrates should be on the same y axis scale to aid in interpretation and comparison (both 60% max).

---

## Author Comment (AC1) · 23 Jul 2019

We thank the reviewer for their comment that the experiments we report are novel and elegant, and that our results are exciting. Below we answer the substantive points that the reviewer raised:

1. The reviewer raises a valid question as to whether it is appropriate to scale up from processes measured in 10cm diameter cores to rates and measurements normalised to per m2. We acknowledge that presenting results as per cm2 is a more conservative approach, however we note that per m2 is the standard normalisation in the rest of the literature. We are happy to provide results normalised to cm2, but feel strongly that the

[Figure]

per m2 version should also be presented to allow readers easy comparison with the wider literature. We leave this at the discretion of the editor.

2. The reviewer notes that it does not make sense to report results as a mean and standard deviation when n=2. We acknowledge that greater replication is certainly desirable, but not always achievable. Indeed, in this case further replication was prevented y availability of cores, incubation equipment, and time at sea. We are happy to alter the way in which results are presented in line with the reviewer comment to avoid use of mean and standard deviation.

3. We will make the changes suggested in the minor comments.

―――――――――――――――――

---

## Author Comment (AC2) · 23 Jul 2019

The reviewer's argument is that the measurements of in situ C fixation that we present have conceptual flaws, due to having been conducted ex-situ. Thus the sediment that we incubated was cut off from its supply of the electron donors which provide the energy for chemosynthesis, and which are presumably sourced from the upwards flux of hydrothermal fluid from deeper in the sediment. We acknowledge this point, but suggest that it may not have been a serious consideration at the sites which we studied, and does not warrant exclusion of all the benthic inorganic C fixation material.

Firstly, the hydrothermal site that we studied (Hook Ridge, in the Bransfield Strait) was

rather mildly hydrothermal. Hence, as has been reported, vent endemic fauna were almost absent (Bell et al., 2016), there was no increase in faunal biomass close to venting, and downcore profiles of alkalinity, nitrate and ammonium were consistent with normal microbial processes (Aquilina et al., 2013). There were indications of of hydrothermal flux in chloride, sulphate and sulphide profiles, which allowed Aquilina et al. (2013) to calculate hydrothermal advection rates of 9-33 cm y-1. At these low advection rates we suggest that there would not have been sufficient time during our ∼60 h experiment for a noticeable depletion in availability of electron donors supplied by hydrothermal fluid.

Secondly, we measured greatest amounts of benthic inorganic C fixation at our non-hydrothermal control site. The methods we used did not allow us to definitively pinpoint the metabolic processes responsible for inorganic C fixation, but the fact that C fixation was maximal at a non-hydrothermal site suggests that it is not, or not always, inherently linked to hydrothermalism. Indeed this is one of our key findings. Therefore, while our ex-situ incubation technique could have resulted in conservative rate measurements at the hydrothermal site, we do not feel that it would be a proportionate response to exclude all the material about benthic inorganic C fixation.

Finally, we note that other reviewers found the material about benthic inorganic C fixation to be interesting, novel, and worthy of publication.

We will add this discussion of a potential artefact from our experimental technique to our discussion.

In addition the reviewer asks for clarification of methods (e.g depths over which PLFAs were measured, and procedure used to determine whether labelling levels were above background). These details will all be added. Further they also make the same point about use of means and standard deviations as reviewer 1. As stated in the reply to reviewer 1, we will alter our presentation of results to avoid use of means and standard deviations.

**References:**

Aquilina, A., Connelly, D. P., Copley, J. T., Green, D. R. H., Hawkes, J. A., Hepburn, L. E., Huvenne, V. A. I., Marsh, L., Mills, R. A., and Tyler, P. A.: Geochemical and Visual Indicators of Hydrothermal Fluid Flow through a Sediment-Hosted Volcanic Ridge in the Central Bransfield Basin (Antarctica), Plos One, 8, 2013.

Bell, J. B., Woulds, C., Brown, L. E., Sweeting, C. J., Reid, W. D. K., Little, C. T. S., and Glover, A. G.: Macrofaunal ecology of sedimented hydrothermal vents in the Bransfield Strait, Antarctica, Frontiers in Marine Science, 3, 2016.

---

## Author Comment (AC3) · 24 Jul 2019

We thank the reviewer for a supportive review.

The reviewer feels that the methods section could provide more detail, and this will be added in line with this and other reviews.

In particular, the reviewer asks for further detail of the dual labelled phytodetritus that was added to the 'algae' treatment, and this will be provided. In addition we will add acknowledgement and discussion of the fact that the phytodetritus used was fresher and more reactive than the particulate organic matter that usually reaches the depth

of our study sites. This is a common feature of most previous experiments of this type, and we acknowledge that it means that the processing rates we report for algal are likely to be maximum rates. Other revisions to our discussion will include accommodating the reviewer's comment on how we discuss the potential impact of experimental temperature on our measured rates. The reviewer also makes a helpful point about potential reasons for the spatial heterogeneity we observed (potentially sporadic nature of venting, and consequent adaptability of benthic communities), and this will be added to the discussion. We will also add discussion as to whether the duration of our experiments limited our ability to observe transfer of fixed 13C to some C pools.

Other details, clarification, and changes to figure formatting that the reviewer requests, for example the proportion of annual flux that algae addition represented, and a brief description of the macrofaunal community, will be added/changed.

---

## Author Response (AR1)

**BG-2019-198 Response To Reviews**

Dear Professor Treude

Thank you for handling our manuscript, and for inviting a revised version. Please see below responses to all reviewer comments, followed by a marked up revised manuscript.

We thank the three reviewers for supportive and constructive comments. Reviewer comments are summarised / provided below, with replies and details of revisions in **bold**.

Yours sincerely,

Clare Woulds

**Reviewer Comment 1**

We thank the reviewer for their comment that the experiments we report are novel and elegant, and that our results are exciting. The reviewer raised the following substantive points:

The reviewer raises a valid question as to whether it is appropriate to scale up from processes measured in 10cm diameter cores to rates and measurements normalised to per $m^2$. **We acknowledge that presenting results as per $cm^2$ is a more conservative approach, however we note that per $m^2$ is the standard normalisation in the rest of the literature. We are happy to provide results normalised to $cm^2$, but feel strongly that the per $m^2$ version should also be presented to allow readers easy comparison with the wider literature. We leave this at the discretion of the editor.**

The reviewer notes that it does not make sense to report results as a mean and standard deviation when n=2. **We acknowledge that greater replication is certainly desirable, but not always achievable. Indeed, in this case further replication was prevented by the availability of cores, incubation equipment, and tha available time at sea. Results are already listed separately for A and B replicate cores in Table 2. Figures 2 and 3 have been re-plotted to avoid the use of mean and standard deviation values, and now match the style of Fig. 5. Mean values have been replaced with ranges in the text describing the experimental results.**

In addition the revierew requested the following minor changes:

Page 1 Line 1: Should read "Benthic carbon fixation…." **corrected**

Page 1 Line 14-15: "There are no previous direct…" This sentence is not required. Please delete the sentence. **We would prefer to keep this sentence to highlight the novelty of our study, and leave this at the discretion of the editor.**

**Style Definition:** Heading 1: Font: 17 pt, Space Before: 0 pt, After: 10.5 pt, Pattern: Clear (White)

**Style Definition:** Heading 3: Outline numbered + Level: 2 + Numbering Style: 1, 2, 3, … + Start at: 1 + Alignment: Left + Aligned at: 0.63 cm + Indent at: 1.27

Page 1 Lines 15-16: Remove paragraph break. **Removed**
Page 1 Lines 21-22: Remove paragraph break. **Removed**
Page 1 Line 22: Revise to: 'Fixation of inorganic C into bacterial biomass was
observed in all cores/sites.' Please revise as suggested. **Added 'sites' to correct**
Page 3 Line 30 – Page 4 Line 85: Throughout the introduction there are many uses
of 'therefore' and 'however'. 90 % of the time these words are superfluous. Please
revise the introduction to make less use of them. **Text revised to reduce incidence**
**of 'however' and 'therefore'.**
Page 3 Line 32. Split this into two sentences. '…dissolved sulphides and methane.
This supports microbes that combine…' **Changed**
Page 4 Line 63-64: Do you have any supporting literature that can be cited to
support this sentence. **The reference supporting this statement (Bernardino et**
**al., 2012) is provided at the end of the following sentence, once the point is**
**fully made.**
Page 4 Line 72: 'On the contrary however' please revise, this is not well phrased.
**Deleted 'however'**
Page 4 Line 77: Delete sub-heading **Deleted**
Page 4 Line 80: Hypotheses should be 'tested' not 'addressed' **Corrected**
Page 5 Line 108-119: This provides a brief summary of the experimental methods.
Please refer to an alternative source as (following…) where a more detailed
description of the method can be found. **The method is not published at greater**
**length elsewhere, and all details have been provided. We are happy to add**
**further details that are requested (such as those relating to C dose which have**
**been added in response to other reviews).**
Page 5 Line 111-112: *Chlorella spp.* phytodetritus would not be representative of the
algal material processed in Antarctic systems. A diatom would have been a more
appropriate choice of 13C-labelled substrate. **We acknowledge this point, and**
**have added text to the method section, as detailed in response to a similar**
**point made by reviewer 3 (see below).**
Page 5 Lines 119-119: Half a core seems to be a very small volume of sediment for
conducting macrobenthic analysis. Given that the size range of macrobenthic fauna
is variable, and species are mobile, is this sample volume appropriate? I get the
impression that you may be missing something significant by only focusing on half a
core for the bacterial and macrobenthic communities. **We acknowledge this point.**
**This is a standard limitation for isotope tracing experiments, from which**
**samples are needed for a range of different analyses. In addition, when**
**conducted in the deep sea, cores tend to be of relatively small diameter (as**
**opposed to 14-25 cm diameter cores which can be used in shallower settings).**
**We stress that we do not attempt to present a macrofaunal survey based on**
**the organisms picked from our experimental cores, as the volume of sediment**
**used would certainly be too small for that purpose.**
Page 6 Lines 151-158: A lot of potential data has been discarded from the PLFAs by
just focusing on four 'bacteria-specific' fatty acids. It would be interesting to see the
full profiles, particularly as the 13C-labelled bicarbonate treatment may reveal some
insight into which PLFAs might be good indicators of microbial carbon fixation. **The**
**PLFA suites are presented in Figure 4, and described and interpreted in**
**section 3.2. We considered the use of PLFA suites carefully during manuscript**
**preparation, and are not confident in drawing further conclusions from them.**

As previously mentioned, I am not content with the use of standard deviations to
describe variation in the data. Where n = 2, you cannot reliably calculate means or
standard deviations. **Figures and text have been amended accordingly.**
Page 7 Line 168: 'In *the* algae addition experiments…' Please revise. **Corrected**

Page 8 Lines 174-181: I think you could potentially offer more insight into the
microbial processes by considering a wider range of PLFAs for each site. Which
PLFA groups showed greatest label uptake? **We appreciate this suggestion.**
**However, the scope to further interrogate the PLFA data has already been**
**carefully considered. We decided that further conclusions cannot be drawn**
**with an acceptable level of confidence.**

Page 8 Line 175: Normally C19:0 is used as a standard in the PLFA analysis which
may explain why it is found in higher concentrations. **Apologies, C19:0 was indeed**
**used as a standard The values in the text and figures have been adjusted to**
**exclude it.**

Page 8 Line 183: Please revise to 'Faunal uptake of added C differed between the
two replicate cores in all experiments…'**Revised**

Page 8 Line 191-192 and Figure 6: Given the small sample size, I am not convinced
that a community level analysis of faunal feeding responses is appropriate.
Differences in faunal uptake are likely to be driven by spatial variability, with common
taxa such as polychaetes heavily overrepresented. This leads to the 'mixed
macrofauna' category essentially consisting of everything except polychaetes. **We**
**agree that the taxonomic resolution of the data is low, but we still feel it is**
**worth reporting the available information on the identities of the organisms**
**responsible for C uptake. We therefore chose to keep this short section, but**
**have prefaced it with the following text to ensure that the limitation is**
**acknowledged:** 'Small size of individuals meant that organisms had to be pooled for
isotopic analysis, limiting the taxonomic resolution of the faunal uptake data.
Although limited in this way, the data show that […]'

Page 8 Line 191 – Page 9 Line 199: In light of the small sample size please don't
refer to dominance either in terms of faunal abundance or feeding responses. It
would be more appropriate to discuss simply which groups were more/less abundant
and exhibited greater/weaker uptake of the 13C-label. **This section has been**
**edited to avoid using the term 'dominant' or 'dominance'.**

Page 8 Line 196- Page 9 Line 199: This last sentence is confusing, please revise
and clarify. **The sentence has been revised to**: 'In addition, meiofaunal organisms
took up $^{13}$C at Middle Sister, and the bicarbonate $^{13}$C that was transferred to
macrofauna at Hook Ridge was mostly observed in amphipod crustaceans.'

There is frequent use of 'therefore' and 'however', please remove these where
possible. **Discussion text has been edited to remove several uses of each.**

Page 9 Lines 202-212: This paragraph is a description of the results. Please revise
to contextualize your findings. **This text has been edited slightly, but we feel that**
**these features of the data need to be pointed out as a foundation for the**
**material that follows.**

Page 9 Lines 220-222: Long sentence, requires broken up. Please revise. **This**
**sentence has been edited, and broken into two.**

Page 9 Lines 223 Page 10 Line 228: Please revise along the lines of "This is
supported by a recent modelling study which suggested that…. (Bell et al 2017b).
Similar results have also been reported from the methane-rich non-hydrothermal
sediments… (Woulds et al., inpress). **The text has been shortened along the lines**
**suggested.**

Page 10 Line 242-248: I am afraid that this is a major flaw in the overall paper. Given
that temperature is critical to microbial metabolism, the current paper is likely to
seriously underestimate the level of carbon fixation. This needs to be made clearer
earlier in the paper. **The extent to which this is a problem remains an open**
**question. Observations of cores on deck strongly suggested that the in situ**
**temperature was substantially lower than that calculated (by estimating**
**cooling of cores during recovery) by Klinkhammer et al. (2001) – i.e. the extent**
**of hydrothermal venting could have changed over time. Unfortunately, due to**
**kit malfunction, we did not have equipment available for in situ sediment**
**temperature measurements. In the absence of this we feel that the**
**experimental approach used here has as much chance of being correct as if**
**we had used the in situ temperature calculated by Klinkhammer et al several**
**years earlier. Considering this, we feel that acknowledgement and discussion**
**of the potential impact of temperature is correctly placed here.**

Page 11 Lines 273-275: Based on two replicates, it would only be possible to
discuss the magnitude of the differences and perhaps compare these between sites.
Remove reference to standard deviations from the discussion. **Reference to**
**standard deviation has been removed.**

Page 12 Line 285: Delete 'rather' **Deleted**

Page 12 Line 298: Revise to '…Branfield Strait. Therefore…' **A slightly different**
**edit has been made in response to an earlier comment.**

Page 12 Lines 298-299: Here you are discussing the effects of temperature on
metabolic rates. Here you should consider the impacts of rate limitation and do a
quick literature search. There is quite a large body of literature on  this topic **It is not**
**entirely clear what point the reviewer would like to see added. However, we**
**have done the suggested search, and have added the following:** 'Both low
temperature and food scarcity have previously been observed to limit metabolic rates
in polar environments (Brockington and Peck, 2001; Sommer and Portner, 2002).'

Page 12 Line 302: Delete 'Thus' **Deleted**

Page 13 Line 324-325: Based on the Q10 effect, metabolic activity increases
logarithmically with temperature. As such, a change of 1oC may be more significant
than you assume. I think this may require further explanation. **We acknowledge this**
**theoretical point. However, previous studies which have examined the impact**
**of temperature on this type of experiment (Moodley et al., 2005; Woulds et al.,**
**2009) have not found an exponential response, so extensive additional**
**discussion may not be well founded. We have edited text to allow for the fact**
**that a 1 degree temperature difference could have accounted for part of the**
**difference.**

Page 13 Line 339: Delete 'and thus high biomass benthic communities.' **Deleted**

Page 13 Line 341: Delete 'Further' and replace 'while' with 'Whilst' **Revised**

Page 13 Line 341-342: The comparison between sites was limited by the size of
each sample (half a core), and lack of replication (n = 2). Your experimental design
does not allow you to make any inferences on faunal patchiness. **We accept this**
**limitation, but have noted earlier that a greater degree of variability between**
**replicates was observed then in other experiments conducted in the same**
**way. We feel that it would be remiss to remove mention of faunal patchiness.**

Page 14 Line 347: You cannot use the term 'significant' as this implies the use of
inferential statistical tests. Please revise. **Apologies. The text has been edited to**
**avoid the use of 'significant' here and elsewhere.**

Page 14 Line 354: Replace 'dominant' with 'main' **Revised**

Page 14 Line 357-358: 'Therefore the hydrothermal site (Hook Ridge) in this study
was not the hotspot of C-cycling that we hypothothesised it would be.' You need to
define what is meant here by a 'hotspot of C-cycling.' Is this referring to
chemosynthetic carbon fixation? **This has been clarified, and now reads '**The
hydrothermal site (Hook Ridge) in this study did not show more rapid C-cycling than
other similar experiments, as we hypothesised it would.'

Page 14 Line 358-359: Delete paragraph break. **We would prefer to keep two**
**separate paragraphs, one for each of the main topics of our manuscript.**

Page 14 Line 362: Delete the final sentence. **We would prefer to keep this final**
**statement as a pointer towards future work.**

**Reviewer Comment 2**

Reviewer 2 argues that the measurements of in situ C fixation that we present have
conceptual flaws, due to having been conducted ex-situ. Thus the sediment that we
incubated was cut off from its supply of the electron donors which provide the energy
for chemosynthesis, and which are presumably sourced from the upwards flux of
hydrothermal fluid from deeper in the sediment. **We acknowledge this point, but**
**suggest that it may not have been a serious consideration at the sites which**
**we studied, and does not warrant exclusion of all the benthic inorganic C**
**fixation material.**

**Firstly, the hydrothermal site that we studied (Hook Ridge, in the Bransfield**
**Strait) was rather mildly hydrothermal. Hence, as has been reported, vent**
**endemic fauna were almost absent (Bell et al., 2016), there was no increase in**
**faunal biomass close to venting, and downcore profiles of alkalinity, nitrate**
**and ammonium were consistent with normal microbial processes (Aquilina et**
**al., 2013). There were indications of of hydrothermal flux in chloride, sulphate**
**and sulphide profiles, which allowed Aquilina et al. (2013) to calculate**
**hydrothermal advection rates of 9-33 cm y$^{-1}$. At these low advection rates we**
**suggest that there would not have been sufficient time during our ~60 h**
**experiment for a noticeable depletion in availability of electron donors**
**supplied by hydrothermal fluid.**

**Secondly, we measured greatest amounts of benthic inorganic C fixation at**
**our non-hydrothermal control site. The methods we used did not allow us to**
**definitively pinpoint the metabolic processes responsible for inorganic C**
**fixation, but the fact that C fixation was maximal at a non-hydrothermal site**
**suggests that it is not, or not always, inherently linked to hydrothermalism.**
**Indeed this is one of our key findings. Therefore, while our ex-situ incubation**

**technique could have resulted in conservative rate measurements at the**
**hydrothermal site, we do not feel that it would be a proportionate response to**
**exclude all the material about benthic inorganic C fixation.**

**We have added the following to discussion section 4.1:**

'Experiments were designed to replicate natural conditions as far as practically
possible, while being limited to shipboard rather than in situ methods. One result of
this is that the sediment contained in cores was detached from the upward flux of
hydrothermal fluid, and the electron donors it supplied. This could have limited
inorganic C fixation, which would have impacted the rates measured at Hook Ridge.
We suggest however that this is not a serious limitation, as Hook Ridge was rather
mildly hydrothermal. Vent endemic fauna were almost absent (Bell et al., 2016),
there was no increase in faunal biomass close to venting, downcore profiles of
alkalinity, nitrate and ammonium were consistent with normal microbial processes,
and hydrothermal advection rates were 9-33 cm $y^{-1}$ (Aquilina et al., 2013). At these
low advection rates we suggest that there would not have been sufficient time during
our ~60 h experiments for a noticeable depletion in availability of electron donors
supplied by hydrothermal fluid.'

In addition the reviewer asks for clarification of methods (e.g depths over which
PLFAs were measured, and procedure used to determine whether labelling levels
were above background). These details will all be added. Further they also make the
same point about use of means and standard deviations as reviewer 1. As stated in
the reply to reviewer 1, we will alter our presentation of results to avoid use of means
and standard deviations.

Line 93 Middle sister is not described in Fig 1 (Three Sisters is described). Off-vent
is also not described in Fig 1, but Off-Axis control is described. Please be consistent
through the text and the figures. **Unfortunately re-drawing the figure is not**
**straightforward, however the caption states that 'off-axis control' is the same**
**as 'off vent', and 'Three Sisters' is the same as 'Middle Sister'.**

Table 1 The authors listed the water temperature of each site, but do you have a
data for characterizing each site in terms of venting activities, such as heat flow
value, H+ or CH4 or Cl concentration of pore water? Can you list up some from
Aquilina et al. 2013?? **We do not have the parameters mentioned by the**
**reviewer for all sites (due to low hydrothermal advection), so do not feel that it**
**would be appropriate to add to Table 1. However, we have added the following**
**note to the method section: '**Porewater geochemistry at Middle Sister and Off-Vent were consistent with microbial processes without influence of hydrothermal activity.
Porewater $NO_3^-$ and $NH_4^+$ profiles were indicative of nitrate reduction, but downcore
declines in $SO_4^{2-}$ and $Cl^-$ were lacking over the ~40 cm depth sampled. In contrast, at
Hook Ridge $SO_4^{2-}$ was depleted by up to 11% compared to seawater, and $Cl^-$ by up
to 7%, allowing calculation of hydrothermal advection of 9-33 cm $y^{-1}$ (Aquilina et al.,
2013).'

Line 115 (If the authors decided not to delete Chemoautotrophic C production
results) To give better idea how much 13C- and 15N were dosed into existing DIC or
ammonium, it is needed to indicate them uM. In the line 158, the authors mentioned
that the added 13C-DIC account for 22%, but this must differ between sites because
venting fluid contains high DIC (5-100mM) than bottom water (_2.1mM). **Estimated**
**concentrations in porewaters of added substrates have been added. Due to**
**weak hydrothermalism at Hook Ridge, alkalinity in the surface sediment there**
**is similar to the other sites (although is higher further downcore, Aquilina et**
**al., 2013), there fore there one estimated value is provided for all sites.**

Line 151 It is totally unclear which depths did authors use for each analysis. For
PUFA, all sediment layers were used or not? For faunal, the authors examined 10
cm or deeper? **This detail has been added (0-1 cm for PLFAs, 0-10 cm for**
**fauna).**

Line 173 More specifically, how much 13C-labeling was determined as detection
limits considering natural variations in d13C values? This must be written in M&M.
**M&M text details that natural isotopic baselines were used for individual**
**PLFAs and faunal taxa. In addition, $^{13}$C uptake was only calculated where the**
**difference exceeded analytical variability. This detail has been added to M&M.**

Line 188 Again, how much 13C-enrichments were regarded as 13C uptake?
"Measurable uptake" sounds like even 1 per mil of d13C differences from background are regarded as uptake. **See reply above.**

Line 206 Please describe this for more detail. Not only analytical precision, but also
variations in background samples in replicate (if available) must be considered,
which sometimes shows _5 per mil of variation. **Replicate background data are not**
**available for PLFAs. For fauna they show variability of usually 1 per mil for**
**each taxon, and enrichment was up to 68 per mil. We have been careful to use**
**only data where we are confident there is an unambiguous and quantifiable**
**enrichment (see earlier responses), and have applied further care in**
**acknowledging the limitations of our study and not over-interpreting the data.**

Line 213 If you measured 13C of PLFAs in different layers, it is worth to put vertical
trends in the graphs. **Due to resource constraints we only have these data for**
**the surface 0-1 cm horizon, otherwise we would certainly plot the data**
**downcore.**

Lines 250 and 252 Probably, "mg" is missing. **Corrected, thank you**

Line 273 It is odd that describing "standard deviation" on samples with 2 replicates.
**In line with other reviewer comments we no longer use standard deviations.**

Figure 7 (If the authors decided not to delete Chemoautotrophic C production
results)

It is a bit confusing to show these graphs together; one with "respiration" but one
without. I understand that the respiration of B cannot be measured because of 13C-
DIC addition, but you need to mention that clearly in the caption.

Line 325 The differences in microbial biomass were less than twice, while those in
respiration rates differed _7 times. So, the microbial biomass is not the only reason.
The authors need to discuss about these fact more carefully.

**Reviewer Comment 3**

We thank the reviewer for a supportive review.

The reviewer feels that the methods section could provide more detail, and this will
be added in line with this and other reviews.

In particular, the reviewer asks for further detail of the dual labelled phytodetritus that
was added to the 'algae' treatment, and this is provided. In addition acknowledge
that the phytodetritus used was fresher and more reactive than the particulate
organic matter that usually reaches the depth of our study sites. This is a common
feature of most previous experiments of this type it means that the processing rates
we report for algal are likely to be maximum rates. **We have added the following**
**text to methods section 2.2:**

'It is recognised that such organic detritus is less degraded than the sinking
photosynthetic material which normally reaches the depths of our study sites. This is
a limitation of the method common to all such experiments in the literature, and
means that rates for processing of added C in 'algae' experiments should be
considered maximal. Further, diatom detritus would have been more representative
of local photosynthetic material, but was unfortunately not available.'

LN 83: "inorganic substrates" to bicarbonate (H13CO3-): **changed**

LN 103: A brief description here of the background macrofauna would be
appropriate. **The following text has been added: '**. Polychaetes were numerically
dominant (41-56%), except at Hook Ridge, which was dominated by peracarids, and
oligochaetes were the next most dominant. Vent endemic fauna were represented by
two species of sigoblinid polychaete; S. *contortum* at Hook Ridge, and *Siboglimun*
*sp*. Elsewhere (Bell et al., 2016a).'

LN 112: Product number for your labeled algal material is needed, CIL does not
appear to sell a marine algal detritus that I could find by searching their site. Be
careful with that description too, as it implies a bit of possible reworking given that
you are working at 1000 m depth. If the material is the dual labeled lyophilized algal
cells then it is really fresh algal material for application at a relatively deep site; you
should discuss this if it is the case. An estimate of what portion of the annual flux this
application represents would be appropriate to give more context to the amount of
material be applied. **The part number has been added, as well as**
**acknowledgement that the material is fresher than that which usually arrives at**
**the site depth (see response to reviewer 2). The following has been added to**
**provide context for the amount of algal C added '**This was equivalent to ~1.6% of
total OC in the surface 1 cm of sediment, or ~9% of annual OC input (Bell et al.,
2017b)'.

LN113: Context for the relative amount of application versus what is already there
and available would be useful so the reader can gauge how large the applications
are versus in situ backgrounds for C and N. **This has been added, please see**
**above.**

LN 116: Describe the sampling intervals, this will help to indicate how many
measurement points your rates are determined off of. **Detail added (sampling**
**every 12 h for 60 h).**

LN 206: Provide a range of PLFA or organic ‰ 13C enrichments to support this
claim. It will be more convincing to readers when presented in that manner. **Detail of**
**the steps we took to ensure the reported uptake was real are given in the**
**methods section. Since enrichments above background were often small, and**
**baseline values themselves varied with taxon and specific PLFA, we feel that**
**providing ranges here will be confusing rather than helpful. We will however**
**provide access to archived data via a DOI, which we refer to in the results**
**section.**

LN 212: I appreciate the candid nature of this statement, it shows a realistic
interpretation of the data given the limited replication built into the study. **We thank**
**the reviewer for this comment.**

LN 216: Provide a reference about the chemosynthetic endosymbionts, also an
indication as to the nature of the symbionts, methane oxidizers or sulfur oxidizers
would be appropriate. **References added, along with the detail that most of the**
**endosymbionts will be doing sulphide oxidation.**

LN 235: "important aspect" Is this because it is minor, but potentially widespread?
Vague as written. **This has been re-worded** '…chemoautotrophic C fixation may be
considerably more widespread than previously thought. It is therefore deserving of
further study so that it can be quantitatively incorporated into our understanding of
the marine C-cycle.'

LN 244: So, this study likely represents minimum rates for chemosynthesis. The
authors should phrase it that way and provide context of what the addition
represented in comparison to normally available substrates. Better to focus on what
your study has actually shown than to speculate that rates would have been higher if
in situ temps were maintained. **We agree with the reviewer, the text has been re-**
**phrased as** 'It is therefore likely that the rates measured here for chemosynthetic
incorporation of labelled bicarbonate are minimal rates.'

LN 250 & 252: 0.24-1.02 and 1.29 both need mg in front of C m-2 d-1, respectively
**Corrected, thank you.**

LN 263: Would it be worth trying to isolate polar lipids from archaeal components
given their slow metabolism and the relatively short time frame of this study? **This**
**was initially an objective of the project. However, background organic**
**geochemical work conducted by colleagues found archaeal lipids at very low**
**concentrations, therefore we were very unlikely to succeed in tracing 13C into**
**them given the volume of sample available.**

LN 268-277: Thank you for addressing the variability observed during tracer studies
relying on bacterial mediation of a substrate! Is it worth talking about reasons for
potential hotspots for both heterotrophy and chemosynthetic processes that are
occurring in this system? I would expect variations in vent flows and sporadic availability of resources to give rise to a community that readily adapts to changing
conditions. **This is a good point. We have added the following statement to the**
**relevant discussion section: '**Fine scale distribution of fauna has been show to
relate to variations in concentrations of species such as sulphide and methane
(Levin et al., 2003), therefore the patchiness observed especially at Hook Ridge is
likely related to spatial and temporal fluctuation in hydrothermal advection.'.

LN 294: Provide percentages from the other studies here so the reader can directly
compare these studies. **These have been added.**

LN 316: Does the time period involved in this incubation matter here? Transfer into
symbiont and then into tube worm may take a bit more time and require a stronger
signal to show up as the tracer is sequentially diluted through the two carbon pools?
**Only fixation by endosymbionts would be required in order for labelled C to be**
**detected in the isotopic signature of siboglinid specimens (no further transfer**
**required, as the symbionts live within the annelid tissues). The rate of that**
**process may have been a factor, and the following has been added along with**
**other caveats: '**…or because experiments were not long enough for uptake by
endosymbionts.'

Figures: Figure 1: state that depth is in meters in figure caption. **Added**

Figure 2 & 3: remove blue outline on bars. Considering the low uptake rates,
consider converting into µg to limit the decimal places. But, you are consistent
throughout currently. **Figures have been re-plotted in line with reviewer 1**
**comments, blue outlines have been removed. The reviewer makes a valid point**
**about decimal places, but we prefer to use mg to maintain comparability with**
**the literature.**

Figure 4: Format the letters for the figures into the actual graphs, hard to interpret as
laid out presently. Also resulted in the splitting of the figure between page 24 and 25.
Both substrates should be on the same y axis scale to aid in interpretation and
comparison (both 60% max). **Letters not added to panels, but this can be done**
**depending on what is preferred by typesetters. The reviewer makes a valid**
**point about using the same y-axis scales, but this is not practical as it will**
**make plots difficult to read – especially panel A (currently on 0-20% scale, so**
**would be very small on a 0-60% scale).**

[revised manuscript text omitted]